# Dynamin-Independent Mechanisms of Endocytosis and Receptor Trafficking

**DOI:** 10.3390/cells11162557

**Published:** 2022-08-17

**Authors:** Chayanika Gundu, Vijay Kumar Arruri, Poonam Yadav, Umashanker Navik, Ashutosh Kumar, Veda Sudhir Amalkar, Ajit Vikram, Ravinder Reddy Gaddam

**Affiliations:** 1Department of Pharmacology and Toxicology, National Institute of Pharmaceutical Education and Research (NIPER), Hyderabad 500037, Telangana, India; 2Department of Neurological Surgery, University of Wisconsin, Madison, WI 53792, USA; 3Department of Pharmacology, Central University of Punjab, Bathinda 151001, Punjab, India; 4Department of Pharmacology and Toxicology, National Institute of Pharmaceutical Education and Research (NIPER), Kolkata 700054, West Bengal, India; 5Department of Internal Medicine, Carver College of Medicine, The University of Iowa, Iowa City, IA 52242, USA

**Keywords:** endocytosis, dynamin, microRNAs, non-dynamin GTPases, calcium

## Abstract

Endocytosis is a fundamental mechanism by which cells perform housekeeping functions. It occurs via a variety of mechanisms and involves many regulatory proteins. The GTPase dynamin acts as a “molecular scissor” to form endocytic vesicles and is a critical regulator among the proteins involved in endocytosis. Some GTPases (e.g., Cdc42, arf6, RhoA), membrane proteins (e.g., flotillins, tetraspanins), and secondary messengers (e.g., calcium) mediate dynamin-independent endocytosis. These pathways may be convergent, as multiple pathways exist in a single cell. However, what determines the specific path of endocytosis is complex and challenging to comprehend. This review summarizes the mechanisms of dynamin-independent endocytosis, the involvement of microRNAs, and factors that contribute to the cellular decision about the specific route of endocytosis.

## 1. Introduction

Endocytosis is an evolutionarily conserved cellular process in all eukaryotic cell types [1]. Biologically, it facilitates the turnover and degradation of plasma membrane proteins, receptors, and lipids that play a vital role in the interaction of cells with their external environment and the uptake of nutrients [2]. Clathrin- and caveolin-mediated endocytosis have been widely studied, and dynamins, key regulatory GTPases, play a significant role in this process [3]. Dynamins are large (100 kDa) GTPases and act as mechanoenzymes. Based on their characteristic domains, they are categorized as dynamin 1 (DNM1), dynamin 2 (DNM2), and dynamin 3 (DNM3) [4]. Dynamins form a helical ring at sites of membrane invagination and stimulate GTPase activity, which subsequently aids in the release of encapsulated cargo to the cell’s interior, a critical step in clathrin-mediated endocytosis [5]. Endocytosis mechanisms depend on dynamins are essential for various biological functions such as sarcolemmal repair, oocyte development, neuronal plasticity, and dendritic outgrowth [6,7,8,9]. Furthermore, endocytosis of cellular nutrients and receptors (e.g., interleukin-2 receptor and epidermal growth factor receptor) depends on dynamins [10,11]. Dynamins and related proteins are also involved in phagosome cleavage and mitochondrial dynamics (fusion/fission) regulation . Mutations in dynamins or abnormal dynamin functions are associated with pathological conditions (e.g., neurodegenerative diseases, cancer, epilepsy, heart failure, Charcot-Marie-Tooth disease, osteoporosis) [12,13,14,15,16,17]. 

Though dynamin’s role in cellular endocytosis has been well studied at the molecular level, recent studies shed new light on dynamin-independent internalization and trafficking of endosomes [18,19]. The key players in these processes include actin and actin-associated proteins, small GTPases such as Arf6 (ADP-ribosylation factor 6), Cdc42 (cell division cycle 42), flotillins, and secondary messengers such as calcium [20,21,22]. The recent advancements in small-molecule inhibitors of dynamin GTPases (e.g., Dynasore and Dyngo) and studies using mutated dynamins demonstrate the biological relevance of dynamin-independent endocytosis [23,24]. Hence, endocytic pathways which operate without dynamin have attracted considerable research interest. Figure 1 illustrates the relationship between dynamin-independent endocytosis mechanisms, GTPase enzymes, and other regulatory proteins involved in endocytic processes. 

In this review, we discuss dynamin-independent endocytosis mechanisms, the role of microRNAs (miRNAs), and factors that determine the endocytosis (dynamin-dependent/independent) mechanisms.

## 2. Dynamin-Independent Endocytosis Mechanisms

### 2.1. The Clathrin-Independent Carrier/Glycosylphosphatidylinositol (GPI)-Anchored Protein (AP)-Enriched Endosomal Compartments (CLIC/GEEC; CG) Endocytosis Pathway

Endocytosis of GPI-APs reflects a unique internalization route. It is clathrin- and dynamin-independent, and the cargo is routed into GPI-AP-enriched early endosomal compartments (GEECs) by integration of uncoated tubulovesicular clathrin-independent carriers (CLICs) emanating directly from the plasma membrane [25]. Small GTPase Cdc42 and GTPase regulators (e.g., GRAF1 and Arf1) are involved in CG endocytosis [26]. Cdc42 cycles between active (GTP-bound) and inactive (GDP-bound) state dynamics are crucial in governing CG endocytosis [27]. The dynamin-independent endocytosis of the CG pathway was experimentally confirmed in Drosophila wing discs [28]. This study proved that endocytosis of the secreted morphogen Wingless (wg) via the CG pathway is critical for interaction with its receptor DFrizzled2 (DFz2) and coordinate signaling during Drosophila patterning [28]. The CG pathway maintains the integrity of the plasma membrane [29]. Internalization of various molecules, including GPI-linked proteins [30], T-cell receptors [31], bacterial exotoxins, and VacA toxin [32], also depends on CG endocytosis. Notably, the CG pathway determines the subcellular localization of dysferlin (a skeletal muscle repair protein) in muscular dystrophy [33]. Under physiological conditions, caveolin-1 or -3 increases the surface retention of dysferlin by promoting its exit from the Golgi complex to the plasma membrane. However, during muscular dystrophy (linked to mutations in the caveolin gene), CG endocytosis promotes dysferlin surface retention by increasing its exit from the Golgi complex [33], which signifies its pathological role. 

### 2.2. Arf6-Mediated Endocytosis

ADP-ribosylation factors (Arfs) are Ras family GTPases that are classified into three families [34]. Class I Arfs (Arf1, Arf2, and Arf3) show 97% sequence similarity and are involved in the secretory pathway and lipid metabolism [35,36]. The function of Class II Arfs (Arf4 and Arf5) is not entirely understood; however, Arf4’s role has been recently identified in the recycling pathway of endosomes [37]. Arf6 belongs to Class III, a relatively more explored member of Arfs, and regulates membrane trafficking and endocytosis [38]. Arf6’s effects on membrane trafficking were first observed by Peters et al. [39]. They identified that the constitutive activity of Arf6 is required for plasma membrane invaginations. Furthermore, dominant-negative Arf6 induces an accumulation of coated-endocytic structures in the rhabdomyosarcoma cells (RD4), suggesting its role in endosomal trafficking [39]. The molecular machinery of Arf6-mediated endocytosis is uncertain but could involve actin remodeling as its activation results in cell surface protrusions [40]. GTP binding and the downstream conformational changes in Arf6 (Arf6-GTP conformation; active state) are crucial for effector recruitment and downstream trafficking. The involvement of Arf6 in dynamin-independent endocytosis was confirmed by multiple studies [41,42,43]. For instance, peptide-loaded major histocompatibility complex II (pMHC-II) molecules in HeLa-CIITA cells, B cells, and dendritic cells internalize through Arf6-mediated endocytosis [41]. In neuroendocrine cells, the internalization of synaptosome-associated protein is also regulated by Arf6 [43]. Likewise, Arf6 mediates endocytosis of multiple proteins and receptors (e.g., MHC-I, IL-2Rα, muscarinic M2 receptors, β1 integrins, PMP22) and is implicated in cellular function regulation [42,44,45,46]. Physiological functions of Arf6-mediated endocytic trafficking include regulation of plasma membrane availability for migratory cells, modification of membrane lipids, and actin cytoskeleton organization [46]. Thus, understanding Arf6 endocytic pathways could pave the way for treating diseases such as cancer [47,48]. 

### 2.3. Flotillins

Flotillins are evolutionarily conserved membrane-associated proteins and include flotillin-1 and flotillin-2. They are ubiquitously expressed in mammalian tissues and regulate many cellular functions ranging from cell signaling, regulation of the cortical cytoskeleton, endocytosis, and protein trafficking to gene expression [49,50]. The distinct feature of flotillins is their ability to co-cluster into discrete microdomains in the plasma membrane, driving compartmentalization and functional specialization within the membrane [51,52]. These microdomains are traversed laterally within the membrane and subsequently bud into the cell, distinct from caveolae and clathrin- and dynamin-dependent endocytosis [53]. The driving force behind the distribution of flotillin microdomains in the membrane and endocytosis is thought to be the phosphorylation of tyrosine residues in flotillins by Fyn (Tyr160 in flotillin-1 and Tyr163 in flotillin-2) and related Src family kinases. The phosphorylation of flotillins by these enzymes results in the internalization of microdomains and trafficking of flotillin proteins from the plasma membrane to late endosomes and lysosomes [54,55,56]. Many proteins undergo flotillin-mediated endocytosis, such as GPI-anchored protein CD59, the receptor for cholera toxin, glycerophospholipid GM1 (monosialotetrahexosyl ganglioside), and teratocarcinoma-derived growth factor 1 (TDGF-1) [57,58]. Flotillin-mediated endocytosis was found to play a crucial role in the recruitment of chemokine receptor (CXCR4) upon chemokine (CXCL12) treatment of T cells, suggesting the role of flotillins in the generation of specialized surface platforms which facilitate immunological receptor signaling and their endocytosis [59]. Flotillin-mediated endocytosis also play a role in cholesterol uptake and absorption in a Niemann-Pick C1-like 1 (NPC1L1)-dependent manner [60]. Furthermore, flotillin-1 aids in the internalization of semaphorin 3A (Sema3A) and its receptors (neuropilin-1, cell adhesion molecule L1, and plexinA4), which regulate LIMK1 activity, actin cytoskeleton dynamics, and adhesion in cortical neurons [61]. Though the flotillins are involved in dynamin-independent endocytosis pathways, few studies argued that dynamin might be involved in the endocytosis of cargo by flotillin-2 [62,63]. The amphibious nature of flotillins in endocytic pathways could be due to their activity as adaptors to enroute specific cargo towards alternative dynamin-dependent endocytic pathways [62]. Further studies are warranted to explore the mechanistic aspects of flotillins’ interaction with specific cargo and vesicle formation during dynamin-independent endocytosis.

### 2.4. Calcium-Mediated Regulation of Endocytosis

Calcium plays a vital role in synaptic transmission and internalization of membrane vesicles following neurotransmitter release via exocytosis [64]. The key findings on calcium endocytosis suggest that calcium threshold is critical in determining the rate of endocytosis, as an initial increase in calcium current triggers slow endocytosis, followed by bulk and rapid endocytosis, and finally, endocytosis overshoot. However, there are a few exceptions to it [65]. An alteration in calcium level is sensed by calcium sensors such as calcineurin (CaN), calmodulin (CaM), and synaptotagmin (syt), which leads to the opening of the voltage-gated calcium channels and initiation of endocytosis [66]. The prolonged calcium influx determines the endocytic capacity in chromaffin cells [67] and hippocampal synapses [68]. Though most of the calcium-regulated receptor endocytosis is clathrin/dynamin-dependent, recent studies have identified that calcium regulates endocytosis in a dynamin-independent manner. For instance, Holstein et al. observed that calcium-sensing receptor (CaSR)-mediated ERK1/2 activation depends on calcium, but neither dominant-negative dynamin (K44A) nor dominant-negative β-arrestin inhibits ERK1/2 activation by CaSR agonists (NPS R-467 or CaCl_2_). These results suggest that CaSR-mediated ERK1/2 signaling occurs in a dynamin-independent manner [69]. Studies also indicate that the dynamin-independent endocytosis processes are regulated by calcium influx, ATP, and pH changes . For example, internalization or desensitization of glutamate receptors in astrocytes or TRPV ion channels depend on ATP and Ca^2+^ influx, respectively. In neuronal cells, the neuronal activity and cell signaling pathways such as PKA activation are regulated by intracellular calcium [70]. A distinct endocytosis process termed massive endocytosis (MEND) is a calcium-dependent and dynamin-independent mechanism and internalizes ordered plasma membrane domains [71,72,73]. Lipidic forces and membrane phase separation are thought to be responsible for the MEND endocytic mechanism [26]. Taken together, calcium-regulated endocytosis plays a role in synaptic transmission, plasma membrane integrity, and divergent cell signaling events mediated by receptors. The significance of calcium at the level of endocytic machinery needs to be further investigated.

### 2.5. Pinocytosis

Pinocytosis is a process of cell drinking where the cell internalizes extracellular fluid via endocytosis [74]. Pinocytosis is further classified into micropinocytosis and macropinocytosis based on the diameter of the vesicle formation and the amount of solute uptake [75]. Multiple studies delineated dynamin-independent micropinocytosis and macropinocytosis [24,76,77]. For example, granzyme B (a proapoptotic enzyme) uptake is mediated through micropinocytosis in K44A-dynamin (mutant dynamin)-overexpressing HeLa cells [76]. A study in porcine aortic endothelial (PAE) cells showed that antibody-induced epidermal growth factor receptor (EGFR) endocytosis is neither sensitive to clathrin nor dependent on dynamin but is inhibited by amiloride and latrunculin B, which are known inhibitors of macropinocytosis [24]. 

Micropinocytosis takes part in several cellular functions. The uptake of thyroglobulin (Tg) occurs through micropinocytosis, where Tg rapidly fuses with primary lysosomes and releases iodinated compounds upon subsequent maturation [78]. It is also involved in fetomaternal transfer where proteins are transferred via micropinocytosis across fetal capillary endothelium [79]. Furthermore, insulin uptake by erythrocytes is carried out by micropinocytosis [80]. Plasma membrane invagination during micropinocytosis is mediated by microtubule and cortical actin filament remodeling and small GTPases of the Rab family, including Rab34 and Rab7 [81,82]. Though recent reports have demonstrated dynamin-independent micropinocytosis, these mechanisms remain to be explored.

Macropinocytosis is an actin-driven endocytic process and it can be either a signal-dependent process (occurs in response to growth factor stimulation) [83,84] or a constitutive form of macropinocytosis [85,86]. Macropinocytosis is involved in the regulation of cell motility, nutrient sensing, tumor progression and metastasis, and chemotactic responses. Constitutive macropinocytosis is common in immune cells (e.g., macrophages, dendritic cells) and stringently requires the presence of extracellular calcium [87,88,89]. For example, extracellular calcium sensed by G-protein-coupled calcium-sensing receptors (CaSR) promotes macropinocytosis through Rac and Cdc42 and regulates the housekeeping function of macrophages. However, constitutive macropinocytosis is not entirely confined to immune cells. Heterogenous expression of CaSR in other cell types also promotes constitutive macropinocytosis. 

### 2.6. Entosis

Entosis is a non-apoptotic cell death program with a cell-in-cell (CIC) cytological feature. It is often referred to as a cannibalistic quality of cells, where those detached from adherent junctions are engulfed by other healthy cells [90]. In contrast to phagocytosis, entosis involves the engulfment of live cells, which may or may not be degraded through lysosomes. This phenomenon is commonly observed in cancer cells and during developmental changes of different tissues. RhoA-family proteins, including RhoA-GTPase, RhoA effector kinases (Rho-kinases I and II; ROCK I/II), and Rho-GTPase-activating proteins, play a crucial role in this phenomenon [90,91]. The oncogenic protein Kras regulates Rac1 activation and increases entosis [91]. This is supported by the observations that the increase in the tumor-associated mutations in TP53, Kras, and MYC positively correlates with entosis in pancreatic ductal adenocarcinoma (PDAC) [92].

Under physiological conditions, entosis prevents the accumulation of unhealthy cells. For instance, entosis prevents aneuploidy in a p53-dependent manner during mitosis. Generation of an abnormal cell or DNA damage during prolonged metaphase in mitosis triggers the activation of tumor suppressor p53, which drives cell-in-cell formation via Rnd3-compartmentalized RhoA activity [93]. Furthermore, mitosis and entosis are regulated by Cdc42 as its loss results in deadhesion, cell rounding, and, consequently, the RhoA activity-initiated entosis [94]. Entosis has also been described as essential for recycling amino acids from the engulfed cells via lysosomal degradation, a process that is majorly regulated by mTORC1 activity [95]. Entosis is stimulated by glucose starvation; the starving cells take up cells with high AMPK activity and replenish required nutrients from the ingested cell [96]. During the establishment of pregnancy, the passage of blastocyst trophectoderm through uterine luminal epithelial cells, and direct physical contact with the underneath stroma is an essential step for embryo implantation. At this stage, trophoblast cells remove luminal epithelial cells via entosis [97]. Though entosis is an integral part of cell physiology, several aspects of the process remain to be explored. A detailed account of entosis and the processes that regulate it will provide an understanding of tumor survival under nutrient-deprived conditions and identify the factors regulating the tumor’s aggressiveness. Figure 2 illustrates the physiological and pathological implications of different dynamin-independent endocytosis mechanisms.

### 2.7. Uncharacterized Dynamin-Independent Endocytic Mechanisms

There are distinct yet-to-be-characterized endocytic mechanisms that are regulated by a dynamin-independent mechanism. For example, a pinocytic chaperone pincher is involved in the internalization of myelin-associated inhibitors for axonal growth Nogo-A in neurites. Nogo-A internalization is associated with neuronal growth arrest via the cAMP pathway [98]. Another study conducted by Haugsten et al. demonstrated dynamin-independent internalization of fibroblast growth factor receptor 3 (FGFR3). Neither mutant dynamin nor Arf6, flotillin 1 and 2, or Cdc42 depletion affect the FGFR3 uptake, indicating a new endocytic mechanism (clathrin- and dynamin-independent endocytosis of FGFR3 implications for signaling) [99]. Thus, exploring the underlying mechanisms of this uncharacterized endocytosis of FGFR3 is essential in certain disease conditions (skeletal muscle disorders and malignancies) where FGFR3 is prominently dysregulated [100]. Spoden et al. observed that tetraspanins, an evolutionarily conserved family of proteins that contain four membrane-spanning domains, including CD63 and CD151, aid in the entry of HPV-16 into human epithelial cells. This entry is independent of classical endocytic pathways mediated by clathrin, caveolin, flotillin, and dynamin [101]. In a recent study, Wesen and colleagues observed that intraneuronal uptake of Aβ (1–42) is independent of clathrin and dynamin, similar to internalization by CLICs or macropinocytosis, but their molecular pathways are unidentified [102]. Intraneuronal accumulation of Aβ (1–42) is an early pathological symptom of Alzheimer’s disease; therefore, understanding mechanisms of intraneuronal uptake of Aβ (1–42) could have important implications for the development of future Aβ-clearing therapies. Altogether, this experimental evidence indicates the pathophysiological importance of these uncharacterized endocytic mechanisms. 

## 3. Implications of miRNAs in Endocytosis Process Regulation

MicroRNAs (miRNAs), also known as small non-coding RNAs, are widely recognized for regulating gene expression, which modulates physiological and pathological processes. However, their role in endocytosis and intracellular trafficking remains largely unknown. Over the past decade, research on miRNAs has shown their involvement in endocytosis processes [103,104,105]. For instance, the miR-199 family (miR-199a-5p and miR-199b-5p) markedly regulates the expression of endocytosis mediators such as clathrin heavy chain (CLTC), Rab5A, LDLR, and caveolin-1 [103]. Another study has shown that miR-133α promotes the recycling of neurotensin receptor 1 (NTR1) without affecting its internalization and stimulates the proinflammatory signaling of neurotensin (NT) [106,107]. In addition, miR-290 promotes fibroblast growth factor–extracellular signal-regulated kinase (FGF-ERK) signaling via actin/dynamin-dependent endocytosis and regulates embryonic stem cell proliferation and differentiation [105]. miRNA’s role in endocytosis is not entirely confined to dynamin-dependent mechanisms. Recent evidence has shown that miRNAs regulate proteins and processes of dynamin-independent endocytosis [108,109,110]. For example, miR-218 is known to affect Arf6 expression and modulate pancreatic ductal adenocarcinoma invasion [110]. Similarly, miR-124 inhibits the flotillin-1 expression and suppresses cell growth and migration in breast cancer cell lines [111], while miR-485-5p and miR-34a inhibit flotillin-2 expression and reduce cell proliferation and migration in cancer cell lines [112,113]. Furthermore, miR-135a downregulates ROCK I/II (critical regulators of entosis) and decreases prostate cancer cell migration and invasion [108]. However, it is unknown whether altered expression of these proteins by miRNAs affects their endocytosis and contributes to the observed phenotype. We recently reported that miR-204 promotes APJ (apelin-receptor) endocytosis via a dynamin-independent, calcium-dependent mechanism and confers protection against cardiac hypertrophy and dysfunction [104]. Others have also reported that miR-regulated calcium-mediated receptor endocytosis is involved in maintaining neuronal synaptic plasticity. For example, miR-24-3p regulates hippocalcin, a calcium sensor protein, which acts as a molecular link between calcium entry into the neurons through N-methyl-D-aspartate (NMDA) receptors and regulation of α-amino-3-hydroxy-5-methyl-4-isoxazole propionic acid receptor (AMPAR) endocytosis. [114,115]. Furthermore, miRNAs have a key biological function in micropinocytosis. For instance, miR-103/107 family members coordinately suppress micropinocytosis in primary human limbal keratinocytes (HLEKs). Loss of miR-103/107 causes dysregulation of micropinocytosis with the formation of large vacuoles, primarily through the upregulation of Src, Ras, and Ankfy1 [109]. miRNAs are not only involved in cellular function through proteins or GPCRs endocytosis but also determine signaling activation and outcome [103,104,105,107]. For example, miR-133α promotes the recycling of NTR1 and regulates neurotensin proinflammatory signaling [106,107]. On the other hand, cardiac miR-204-5p favors dynamin-independent-calcium mediated APJ endocytosis over dynamin-dependent endocytosis and improves cardiac dysfunction [104]. Together, miRNA’s role in endocytosis regulation is emerging. miRNAs not only regulate proteins of endocytosis but also affect their endocytosis processes and determine functional outcome, however, further research is warranted concerning the functional significance of miRNAs in endocytosis and their therapeutic implications in different diseases. Figure 3 illustrates the involvement of miRNAs in the regulation of different dynamin-independent endocytosis. 

## 4. Pathophysiological Role of Dynamin-Independent Endocytosis and Therapeutic Implications

Dynamin-independent endocytosis mechanisms are widely implied to regulate physiological functions and pathological processes. Several nutrients and cellular components are internalized through endocytosis and affect cellular physiology. For example, Arf6-mediated endocytosis is crucial for nutrient and ion transport, cell–cell interactions (Glut1, potassium channels, mucolipin-2, CD98, and CD147), and immunological functions (MHCI, Cd1a, and MHCII) [116]. Folate receptor endocytosis by Cdc42 is essential for nutrient balance [30]. Maintaining plasma membrane tension is vital for various cellular processes, and CG endocytosis is implied in maintaining it . Non-dynamin GTPases (such as Rac1 and RhoA) regulate cell polarity and transcytosis of essential proteins [117,118,119]. It is noteworthy that the endosomal system plays a role in initiating and propagating intraneuronal signaling events and guides axonal growth, dendritic branching, and survival efforts [120]. Furthermore, neurotrophin-stimulated tyrosine receptor kinase (TRK) endocytosis is mediated by macropinocytosis and ensures proper intraneuronal signaling cascades [121]. Dynamin-dependent endocytosis has also been linked to the pathogenesis of several diseases. For instance, a vast number of endocytic proteins in nasal epithelial cells favor the entry of SARS-CoV-2 [122]. Several theories have linked the dysregulated endocytic process with early-stage cancers (disruption of cell–cell junctions and loss of morphological polarity) [123]. The dynamin-independent endocytosis is involved in modulating cell signaling and migration linked with the pathogenesis of several diseases, including cancer and autoimmune disorders [124,125]. Together, these studies signify the pathophysiological role of dynamin-independent endocytic mechanisms. 

Endocytosis is a complex biological trafficking process within cells that attracts considerable therapeutic interest, as dysregulation of endocytosis has been linked to the pathogenesis of several diseases. In astrocytes, the dynamin-independent endocytosis by Rab5 significantly upregulated an amyloid-β-peptide and thus can be further explored as a therapeutic target in Alzheimer’s disease [79]. Similarly, Cdc42-mediated endocytosis of folate receptors can be exploited for developing therapeutic strategies for cancer and inflammatory diseases [35]. SARS-CoV-2 utilizes distinct endocytic pathways to enter respiratory epithelial cells of humans, and targeting these mechanisms might restrict viral entry [122]. Hence, future studies are warranted in this area to combat devastating infectious diseases such as COVID-19. A recent discovery by Renard HF et al. showed that endothelin-A3, upon interaction with galectin-8, modulates the abundance of CD166-containing early endocytic carriers at the cell surface and thereby regulates cell–cell adhesion and metastatic properties of cancer cells [126]. Thus, developing strategies to target this uncharacterized endocytosis restricts the metastasis process of cancer. Additionally, studies have demonstrated the involvement of Rac in the regulation of viral entry in target cells. The measles virus (MeV), which is responsible for respiratory disease and acts as an oncolytic agent, gains entry into the cell via the PVRL4-mediated macropinocytosis pathway involving Rac1 [127]. Hence, understanding molecular mechanisms of different dynamin-independent endocytic pathways in the context of etiopathogenesis of these diseases could yield a suitable target identification, thereby fostering the drug discovery process in the endocytosis arena, a relatively less explored field by biologists. 

The dynamic plasma membrane hosts numerous endocytic pathways. While existing data suggest distinct molecular machinery and their cargo specificity in internalization and trafficking during endocytosis, several questions remain unanswered regarding its diversity, complexity in molecular machinery, and choice of internalization route in a divergent cellular context. The endocytosis fate of cargo cannot be generalized, and it can be varied depending on the type of internalization, extracellular environment of the cell, pathophysiological status, cell type, receptor subtype, etc. In the context of receptor trafficking, there have been several instances where a single receptor can undergo internalization via both dynamin-dependent and independent pathways. For example, internalization of EGFR through a clathrin-dynamin-dependent mechanism recycles the receptor to the plasma membrane and prolongs EGFR activity, which may have a role in cell proliferation and cancer cell growth [128]. In contrast, EGFR internalization through dynamin-independent endocytosis directs the receptor towards lysosomal degradation, inhibiting EGFR effects [11]. Thus, potentiating dynamin-independent uptake of EGFR can be useful in mitigating cell proliferation in cancer cells. In a similar example, platelet-derived growth factor receptor β (PDGFRβ) undergoes dynamin-dependent as well as independent endocytosis [129], and RhoB and Cdc42 are involved in the dynamin-independent internalization of PDGFRβ [130]. Internalization of PDGFRβ via a dynamin-dependent pathway fully activates STAT3, increasing MYC expression and stimulating DNA synthesis during cell proliferation. Inhibiting dynamin-dependent PDGFRβ internalization results in impaired STAT3 pathway activation, leading to hampered cell proliferation. In contrast, dynamin-independent uptake of PDGFRβ does not completely activate STAT3 signaling [131]. As STAT3 signaling is essential for cell proliferation, fibrosis, and cell migration, further in-depth analysis of dynamin-independent uptake of PDGFRβ and its effect on STAT3 signaling would provide novel insights into various diseases. Another example includes the internalization of nicotinic acetylcholine receptor (nAChR), where nAChR undergoes Rac1-mediated endocytosis upon cholesterol binding. However, in the absence of cholesterol, Arf6 aids in nAChR endocytosis [132]. The subcellular localization of dysferlin is determined by disease status [33]. Under physiological conditions, caveolin-1 or -3 increases the surface retention of dysferlin, whereas during muscular dystrophy, CG endocytosis promotes its surface retention by increasing its exit from the Golgi complex [33]. Moreover, the type of receptor internalization varies in different subtypes of receptors belonging to the same family. For example, dopamine receptors D1 and D2 undergo distinct endocytosis processes, where the D1 receptor is internalized through a dynamin-dependent mechanism and the D2 receptor undergoes dynamin-independent endocytosis and mediates the early endosome to lysosomal degradation [133]. Together, studies clearly indicate the importance of dynamin-independent endocytic mechanisms in different pathophysiological status, signaling activation, and outcome. Further understanding of these endocytic pathways in different diseases leads to the therapeutically attractive targets to treat disease conditions. Table 1 describes the list of receptors and proteins that undergo different endocytic mechanisms to regulate physiological and pathological functions.

Collectively, dynamin-independent endocytic mechanisms play important roles in multiple processes ranging from physiological functions to pathological conditions. Of note, unraveling the pathways that intersect dynamin-dependent and -independent endocytosis is challenging and often leads to a common molecular outcome. The use of advanced molecular tools involving single-cell genomic studies and characterization of the endocytic pathway in the context of specific cargo being routed and their trafficking could provide helpful insights in dissecting the dynamin-independent endocytic biology at the molecular level. Furthermore, miR’s role in endocytic regulation is emerging, and determining their role in signaling and the functional outcome has therapeutic importance. Taken together, understanding these mechanisms of endocytosis might offer disease-modifying strategies and serve as a valuable tool for designing novel therapeutic candidates for such diseases.

## Figures and Tables

**Figure 1 cells-11-02557-f001:**
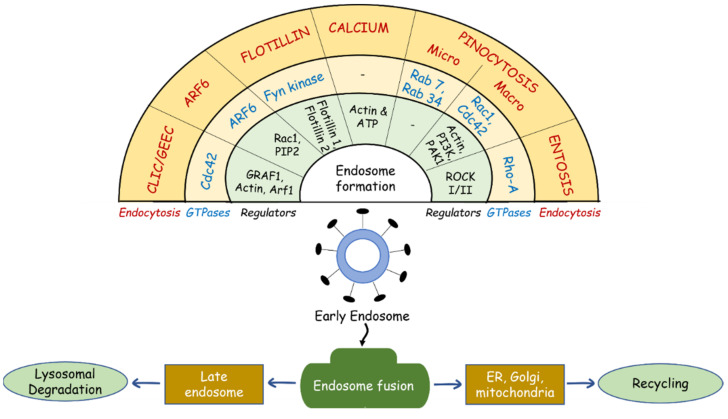
Multiple endocytic mechanisms and their unique GTPases and regulators. CLIC/GEEC, Clathrin-independent carrier/glycosylphosphatidylinositol (GPI)-anchored protein (AP)-enriched endosomal compartments; CDC42, cell division cycle 42; GRAF1, GTPase regulator associated with focal adhesion kinase 1; Arf1, ADP ribosylation factor 1; Arf6, ADP ribosylation factor 6; Rac1, Rac family small GTPase 1; PIP2, phosphatidylinositol-4,5-bisphosphate; Fyn kinase, Src family tyrosine kinase; ATP, adenosine triphosphate; Rab 7, 34, member RAS oncogene family; Rho-A, Ras homolog family member A; ROCK I/II, Rho-kinases I and II.

**Figure 2 cells-11-02557-f002:**
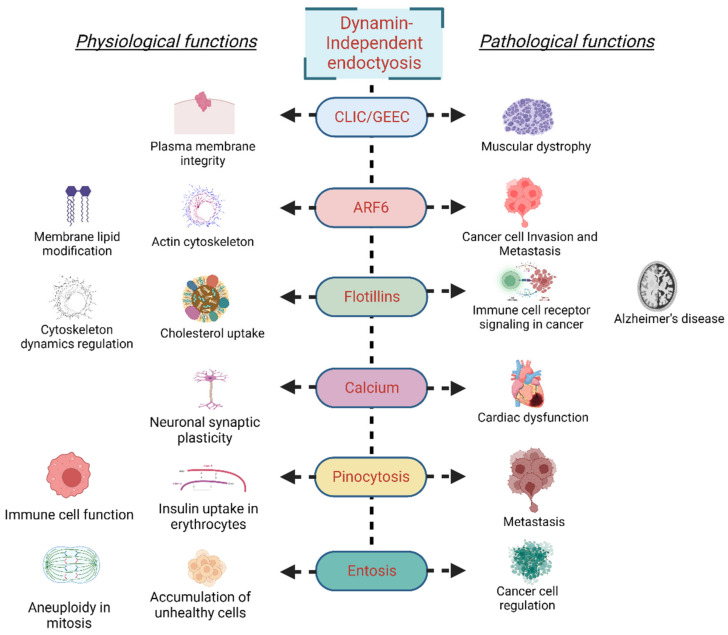
Physiological and pathological implications of different dynamin-independent endocytosis mechanisms.

**Figure 3 cells-11-02557-f003:**
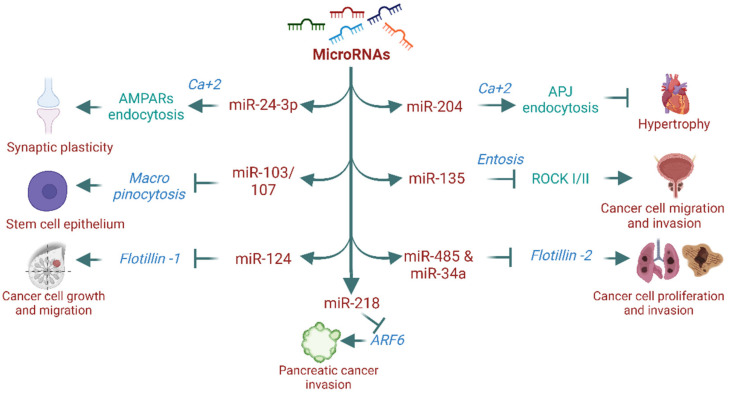
MicroRNA-mediated regulation of proteins/processes of dynamin-independent mechanisms.

**Table 1 cells-11-02557-t001:** A list of receptors and proteins that undergo different endocytic mechanisms to regulate physiological and pathological functions.

Receptor/Protein	Endocytosis		Function	References
	Dynamin Dependent	Dynamin Independent		
Pathological status				
Dysferlin	Caveolin-1 (Cav1)		Increases dysferlin plasma membrane retention	[33]
		CLIC/GEEC	Promotes rapid exit of dysferlin from Golgi complex to the plasma membrane during muscular dystrophy	
**Signaling outcome**				
Apelin receptor (APJ)	Clathrin		Promotes cardiac hypertrophy and dysfunction	[134]
		Calcium	Prevents cardiac hypertrophy and dysfunction	[104]
**Signaling activation**				
Wingless (wg)	CLIC/GEEC		wg containing CLIC/GEEC endosomes fuse with DFz2 containing clathrin endosomes to initiate wg signaling, which is necessary for wing discs signaling in Drosophila	[28]
DFrizzled2 (DFz2, receptor for wingless)		Clathrin		
ATP-binding cassette transporter 1 (ABCA1)	Dynamin-2		Contributes to ABCA1 recycling and enhances efflux of intracellular cholesterol	[135]
		ARF6	Responsible for internalization of ABCA1, leading to its degradation. This pathway is important in the regulation of ABCA1 abundance and efflux of plasma membrane cholesterol	
Low-density lipoprotein receptor (LDLR)	Clathrin		Promotes LDL-LDLR internalization and signaling	[136]
		Epsin	Enhances the lysosomal degradation of LDLR	
**Receptor subtype**				
Dopamine receptors (DRD1 and DRD2)	Clathrin		Involved in endocytosis of DRD1 receptor and its signaling	
		Tetraspanin-7 (TSPAN7)	Promotes endocytosis of DRD2 receptor and regulates its functional activity	[137]
Fibroblast growth factor receptors (FGFR1 and FGFR3)	Clathrin		Promotes FGFR1 endocytosis-faster endocytosis rate	[99]
		Unknown	Promotes FGFR3 internalization-slower endocytosis rate	

## Data Availability

Not applicable.

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
