# Peer review of "Dynamin-Independent Mechanisms of Endocytosis and Receptor Trafficking"

_cells, 2022, doi:10.3390/cells11162557_

Round 1
Reviewer 1 Report
In their manuscript entitled "Dynamin-independent mechanisms of endocytosis and receptor trafficking" Gundu et al. review the current literature on dynamin-independent endocytosis pathways and their physiological significance. This is a timely review on a broadening field and in principle could help to structure it. The manuscript, however needs to be thoroughly reworked to make it useful for the reader.
Comments on specific points
1. In the chapter on uncharacterized dynamin-independent endocytic mechanisms, the authors describe the uptake of Il-2Rb as an example: they are mistaken in this respect. In the original paper by Lamaze et al. (2001) it has been shown that this process is dynamin-dependent and an involvement of the dynamin interaction partner cortactin has been demonstrated by others (Grassart et al. (2008) EMBO Rep. 9, 356-362). The authors should carefully check all their claims with what has been published in the literature to avoid losing the confidence of their readership.
2. Ca2+- dependent endocytosis: there is also a type of constitutive macropinocytosis, active in dendritic cells and macrophages, that depends on extracellular Ca2+ detected by Ca2+ sensing receptors (CaSR). Heterologous expression of these of receptors confers the ability to drive this special type of macropinocytosis in cell types that were not able to support it (for reference see: Canton et al. (2016) Nat. Commun. 7, 11284. The processes the authors discuss may in fact be identical to this type of constitutive macropinocytosis. The authors should check the respective publications and discuss them in this chapter.
3. I do not understand why the authors have excluded macropinocytosis from their review. This is also dynamin-independent and therefore would have been a candidate. They may change this in conjunction with comment 2 and also change Fig. 2 accordingly.
4. The figures are not embedded in the text at all but remain free-floating. There needs to be an anchoring of the figures and they need to be discussed in the text. What is the point of showing figures if you do not use them for your arguments?
5. Fig. 1 needs to be improved: on the fast recycling pathway cargo is depicted freely in the cytosol, a surrounding membrane is missing; What is the meaning of the double red line around the plasmamembrane invagination? Is this meant to be a coat? Inconsistent/confusing representations of membranes: sometimes depicted with a double line (phospholipid bilayer), sometimes as a single line only: the representations need to be consistent throughout and not erratic. Furthermore, an explanation for the changing symbols for endolysosomal content is missing! Some of these symbols reappear in the ER/Golgi region- are these vesicles? The same symbols are used for intraendosomal structures. Please make this figure consistent, because as it is now, it leaves the reader confused!
6. The English in the text is not good: often strange or blatantly wrong wording is used, e.g. "the controversy of this long-standing debate was made easy by..." (page 5, 1st. paragraph) or "an accelerated level of calcium" instead of "an elevated level of... ". Sometimes inappropriate exaggerations are used like "dynamins play an inevitable role in endocytosis". Why then dynamin-independent pathways? It is not just language but this concerns the way scientific papers are written: one feels a bit reminded of the first draft of a bachelor thesis. There are more authors than one. Haven`t the others read the manuscript?
7. The citations should stick to one style. The authors have made a mess by citing journals with the full name, with abbreviated name, with large and small caps mixed and so on. Please read the author instructions and follow them!
Author Response
Response to Reviewer 1 Comments
Q1: In their manuscript entitled "Dynamin-independent mechanisms of endocytosis and receptor trafficking" Gundu et al. review the current literature on dynamin-independent endocytosis pathways and their physiological significance. This is a timely review on a broadening field and in principle could help to structure it. The manuscript, however, needs to be thoroughly reworked to make it useful for the reader.
Response: Thank you for finding this review timely. The manuscript was extensively revised in light of the reviewer's suggestions.
Q2: In the chapter on uncharacterized dynamin-independent endocytic mechanisms, the authors describe the uptake of Il-2Rb as an example: they are mistaken in this respect. In the original paper by Lamaze et al. (2001) it has been shown that this process is dynamin-dependent and an involvement of the dynamin interaction partner cortactin has been demonstrated by others (Grassart et al. (2008) EMBO Rep. 9, 356-362). The authors should carefully check all their claims with what has been published in the literature to avoid losing the confidence of their readership.
Response: We regret our error and agree that dynamin-interaction partner cortactin is involved in IL-2Rβ endocytosis. The error is corrected in the revised manuscript.
We also thoroughly checked the manuscript for all such claims and revised it accordingly.
Q3: Ca2+- dependent endocytosis: there is also a type of constitutive macropinocytosis, active in dendritic cells and macrophages, that depends on extracellular Ca2+ detected by Ca2+ sensing receptors (CaSR). Heterologous expression of these of receptors confers the ability to drive this special type of macropinocytosis in cell types that were not able to support it (for reference see: Canton et al. (2016) Nat. Commun. 7, 11284. The processes the authors discuss may in fact be identical to this type of constitutive macropinocytosis. The authors should check the respective publications and discuss them in this chapter.
Response: Thank you very much for this suggestion. We carefully reviewed the calcium-dependent endocytosis section (section 2.4). The processes that fall into the calcium-dependent constitutive macropinocytosis were moved to the section 2.5 where we discussed constitutive macropinocytosis.
Q4: I do not understand why the authors have excluded macropinocytosis from their review. This is also dynamin-independent and therefore would have been a candidate. They may change this in conjunction with comment 2 and also change Fig. 2 accordingly.
Response: Macropinocytosis was discussed under pinocytosis (please see section 2.5) and figure 2 (now Figure 1) revised accordingly.
Q5: The figures are not embedded in the text at all but remain free-floating. There needs to be an anchoring of the figures and they need to be discussed in the text. What is the point of showing figures if you do not use them for your arguments?
Response: The figures were embedded and discussed in the text as the reviewer suggested.
Q6: Fig. 1 needs to be improved: on the fast-recycling pathway cargo is depicted freely in the cytosol, a surrounding membrane is missing; What is the meaning of the double red line around the plasma membrane invagination? Is this meant to be a coat? Inconsistent/confusing representations of membranes: sometimes depicted with a double line (phospholipid bilayer), sometimes as a single line only: the representations need to be consistent throughout and not erratic. Furthermore, an explanation for the changing symbols for endolysosomal content is missing! Some of these symbols reappear in the ER/Golgi region- are these vesicles? The same symbols are used for intraendosomal structures. Please make this figure consistent, because as it is now, it leaves the reader confused!
Response: Thank you very much for this suggestion. After though revision of this manuscript based on the reviewers' comments, we realized that it is not much helpful or advances the understanding of dynamin-independent endocytosis processes, Thus, we removed figure 1 in the revised manuscript
Q7: The English in the text is not good: often strange or blatantly wrong wording is used, e.g. "the controversy of this long-standing debate was made easy by..." (page 5, 1st. paragraph) or "an accelerated level of calcium" instead of "an elevated level of... ". Sometimes inappropriate exaggerations are used like "dynamins play an inevitable role in endocytosis". Why then dynamin-independent pathways?
Response: The manuscript was edited by native english speaker and thoroughly revised for english language, scientific terminology, and typo errors.
Q8: The citations should stick to one style. The authors have made a mess by citing journals with the full name, with abbreviated name, with large and small caps mixed and so on. Please read the author instructions and follow them!
Response: The references style was revised according to the Cells MDPI journal guidelines.

Reviewer 2 Report
Comments to the Authors
In this review, authors try to summarize the different dynamin-independent endocytosis mechanisms and discusses the emerging ideas showing the functional significance of these pathways.
Overall the aim of authors seems to be achieved, however this reviewer still have several concerns.
1) Although authors wrote “the key players in dynamin-independent endocytosis mechanisms include actin and actin-associated proteins, small GTPases like Arf6 (ADP-ribosylation factor 6), Cdc42 (cell division control protein 42), flotillins, and secondary messengers like calcium”, it is still not clear how these processes were revealed to be dynamin-independent.
Especially, in the section for Arf6 mediated endocytosis, it is not refereed to dynamin-independent processes.
This reviewer feels it would be better to add the texts how these process were revealed to be dynamin-independent.
2) The style of cited references should be re-checked.
For several examples are followings.
21 is J Cell Sci but 50 is Journal of cell science.
30 is Journal of Cell Biology but 33 is The Journal of cell biology and 110 is J Cell Biol?
29 is Cell but 37 is cell.
111 is The FEBS journal but 130 is Faseb j.
Please correct all.
3) I was informed following messages.
Thus I read first un-revised version.
Thursday, June 30, 2022 3:56 PM: However, authors decided to withdraw this manuscript as they realized that some important points are missing in current version. The revised version will be provided until tomorrow, so I would like to draw your attention to check the newest version of manuscript before sending your report. I will inform you as soon as we receive it.
July 6 (Japan) : Unfortunately, we still haven't received the revised version from the authors. However, you may just upload the review report for a previous version on the following link.
Author Response
Response to Reviewer 2 Comments
Q1: In this review, authors try to summarize the different dynamin-independent endocytosis mechanisms and discusses the emerging ideas showing the functional significance of these pathways. Overall, the aim of authors seems to be achieved, however this reviewer still has several concerns.
Response: Thank you for the kind words.
Q2: Although authors wrote "the key players in dynamin-independent endocytosis mechanisms include actin and actin-associated proteins, small GTPases like Arf6 (ADP-ribosylation factor 6), Cdc42 (cell division control protein 42), flotillins, and secondary messengers like calcium", it is still not clear how these processes were revealed to be dynamin-independent. Especially, in the section for Arf6 mediated endocytosis, it is not refereed to dynamin-independent processes. This reviewer feels it would be better to add the texts how these processes were revealed to be dynamin-independent.
Response: We included literature in the revised manuscript on how these endocytosis processes were revealed as dynamin-independent.
Q3: The style of cited references should be re-checked. For several examples are followings. 21 is J Cell Sci but 50 is Journal of cell science. 30 is Journal of Cell Biology but 33 is The Journal of cell biology and 110 is J Cell Biol? 29 is Cell but 37 is cell.111 is The FEBS journal but 130 is Faseb j. Please correct all.
Response: The references style was revised according to the Cells MDPI journal guidelines.
Q4: We are sorry for the reviewer's delay in submitting the revised manuscript.
In addition to the original content, we include a new section focusing on microRNAs, as it makes it more suitable for the collection of Cells and also remains yet unsummarized topic in the context of endocytosis.

Round 2
Reviewer 1 Report
Gundu et al. have now improved and streamlined their manuscript and added a chapter on micro RNAs in endocytosis regulation to align their manuscript with the theme of the special issue. They also have reorganized and improved the figures and conceptually embedded them into the text so that the graphics becomes useful for the reader. My only minor point of criticism concerns table 1, which the authors have extended but also changed into a more condensed format that is perhaps somewhat less clear than the previous one.